# Lipid Oxidation Inhibition Capacity of 11 Plant Materials and Extracts Evaluated in Highly Oxidised Cooked Meatballs

**DOI:** 10.3390/foods8090406

**Published:** 2019-09-12

**Authors:** Stina C. M. Burri, Kajsa Granheimer, Marine Rémy, Anders Ekholm, Åsa Håkansson, Kimmo Rumpunen, Eva Tornberg

**Affiliations:** 1Department of Food Technology Engineering and Nutrition, Lund University, Naturvetarvägen 12, 223 62 Lund, Sweden; kajsagranheimer@hotmail.com (K.G.); marine.remy@agrosupdijon.fr (M.R.); asa.hakansson@food.lth.se (Å.H.); eva.tornberg@food.lth.se (E.T.); 2Department of Plant Breeding, Swedish University of Agricultural Sciences, Fjälkestadsvägen 459, 291 94 Kristianstad, Sweden; anders.ekholm@slu.se (A.E.); kimmo.rumpunen@slu.se (K.R.)

**Keywords:** natural antioxidant, phenol, malondialdehyde, processed meat, Folin-Ciocalteu

## Abstract

The underlying mechanism(s) behind the potential carcinogenicity of processed meat is a popular research subject of which the lipid oxidation is a common suspect. Different formulations and cooking parameters of a processed meat product were evaluated for their capacity to induce lipid oxidation. Meatballs made of beef or pork, containing different concentrations of fat (10 or 20 g 100 g^−1^), salt (2 or 4 g 100 g^−1^), subjected to differing cooking types (pan or deep frying), and storage times (1, 7, and 14 days), were evaluated using thiobarbituric reactive substances (TBARS). The deep-fried meatball type most susceptible to oxidation was used as the model meat product for testing the lipid oxidation inhibiting capacity of 11 plant materials and extracts, in two concentrations (100 and 200 mg kg^−1^ gallic acid equivalent (GAE)), measured after 14 days of storage using TBARS. Summer savory lyophilized powder was the most efficient plant material, lowering lipid oxidation to 13.8% and 21.8% at the 200 and 100 mg kg^−1^ concentration, respectively, followed by a sea buckthorn leaf extract, lowering lipid oxidation to 22.9% at 100 mg kg^−1^, compared to the meatball without added antioxidants. The lipid oxidation was thus successfully reduced using these natural antioxidants.

## 1. Introduction

Since the International Agency for Research on Cancer (IARC) released their monograph stating that consumption of red and processed meats are linked to colorectal cancer (CRC) in 2015, research regarding this topic has increased, particularly since the mechanisms leading to these links remain partly unknown [1]. The most commonly proposed factors underlying the link between consumption of red/processed meat and CRC have been attributed to the following, partly overlapping, mechanisms: (I) An increase in oxidative or N-nitrosation load leading to lipid oxidation and DNA adducts in the intestinal epithelium, respectively; (II) stimulation of proliferation of the epithelium by heme or other food metabolites acting either directly or following conversion, e.g., heterocyclic amines (HCAs) and polycyclic aromatic hydrocarbons (PAHs) through high-temperature cooking; and (III) pro-malignant processes triggered by a higher inflammatory response, e.g., by a process where N-glycolylneuraminic acid (Neu5Gc) is involved in developing xenosialitis, an inflammatory syndrome inducing cancer formation and progression [2,3].

Processed meat refers to products typically made of red meat that have undergone curing, salting, or smoking, often containing high amounts of fatty tissues together with endogenous phospholipids [4]. These factors, together with high-temperature cooking, make them susceptible to oxidative reactions, which may contribute to health hazards [5]. The meatball dish, a minced meat product, is one of the Swedish trademarks in traditional cuisine. The world’s largest furniture retailer, IKEA, claims they sell more than one billion meatballs per year [6]. Meatballs are industrially deep-fried, while usually pan-fried when homemade, and are typically made of pork and/or beef meat (fat content ranging between 10 and 20 g 100 g^−1^), onion, egg, breadcrumbs, and salt (ranging between 2 and 4 g 100 g^−1^). Meatballs were chosen as the study material due to the large market and the fact that the formulation and cooking process covers most of the main components of processed meat products in general.

The addition of antioxidants to meat, meat products, and meat model systems has been widely studied for oxidation preventing purposes [7]. Synthetic antioxidants, such as butylated hydroxytoluene (BHT), butylated hydroxyanisole (BHA), propyl gallate (PG), and tertiary butylhydroquinone (TBHQ), are commonly used in the food industry for oxidation-inhibiting purposes but are decreasing in use due to their suspected genotoxicity [5,7]. Hence, there is an increased demand of natural antioxidants, which could prevent the oxidation process of different meat products, potentially decreasing the negative health effects of processed meat products, as well as prolonging the shelf-life and promoting sustainability. For an evaluation of the lipid oxidation-inhibiting capacity of antioxidants in meat and meat products, the thiobarbituric reactive substances (TBARS) assay is frequently used [8,9,10]. Moreover, to obtain reliable results, it is important that the meat model used is appropriate for the product and supplements tested.

The aim of the present study was (I) to develop a relevant oxidized processed meatball model to study the effects of supplemented antioxidants, and (II) investigate lipid oxidation in meatballs without and with a range of plant materials and extracts at different concentrations.

## 2. Materials and Methods

### 2.1. Plant Materials and Extracts

Eleven plant materials and extracts were collected from Denmark, Estonia, Finland, Latvia, and Sweden (Table 1), i.e., from partners of the EU-project “Sustainable plant ingredients for healthier meat products-proof of concepts”. Finnish phenol-rich extracts were prepared using pressurized hot water extraction (PHWE). Samples were obtained using a Dionex ASE 350 accelerated solvent extractor (Thermo Fischer Scientific Inc., Waltham, MA, USA). Extraction temperatures were 110 °C and 120 °C for sea buckthorn (*Hippophae rhamnoides* L.) leaves and bilberry leaves (*Vaccinium myrtillus* L.), respectively. The static extraction time was 1 min for both samples. After PHWE, the extracts were filtered and lyophilized. Estonian phenol-rich extracts were prepared using a pilot-scale solid-liquid Naviglio extractor (Atlas Filtri, Limena, Italy) with 20 mL 100 mL^−1^ ethanol (aq). Extracts of rhubarb root (*Rheum rhabarbarum* L.) and black currant (*Ribes nigrum* L.) leaves were concentrated to half their volume and were lyophilized to a powder. The ethanol of the Swedish extracts (Table 1) was evaporated before they were diluted to the wanted concentrations. The summer savory (*Satureja hortensis* L.) powder was added non-extracted to the meat batter with the same amount of tap water as the extracts. The extracts were dissolved in MilliQ water prior to dilution in tap water and addition to the meat batter. All samples were analyzed for their total phenols content using Folin-Ciocalteu reagent [11].

### 2.2. Chemicals

2-Thiobarbituric acid ≥98 g 100 g^−1^ (TBA), 1,1,3,3-tetramethoxypropane 99 mL 100 mL^−1^ (TMP), trichloroacetic acid ≥99.0 g 100 g^−1^ (TCA), and ethanol 96 mL 100 mL^−1^ were obtained from Sigma-Aldrich Inc., St. Louis, MO, USA. Hydrochloric acid (40 mM L^−1^) and 85 mL 100 mL^−1^ ortho-phosphoric acid (H_3_PO_4_) were obtained from Merck KGaA, Darmstadt, Germany.

### 2.3. Preliminary Trial for Meatball Model Selection

Meatballs were produced in triplicates with two concentrations of NaCl (2 and 4 g 100 g^−1^), fat (10 and 20 g 100 g^−1^) from two types of meat (pork and beef obtained from Atria Sverige AB) and were either pan-fried or deep-fried, and stored for 1, 7, and 14 days (Figure 1). Lean meat from pork shoulder (*Musculus trapezius*) (fat content 4.3 g 100 g^−1^) was adjusted to fat contents of 10 and 20 g 100 g^−1^, respectively, during mincing using pork belly (*M. external abdominal oblique*) (fat content 28.2 g 100 g^−1^). Lean meat from boneless beef knuckle (*M. semitendinosus)* (fat content 5.5 g 100 g^−1^) was adjusted to the same fat contents using a mixture of cuts from beef chuck (*M. deltoideus*) and clod (*M. latissimus dorsi, M. trapezius*, *M. serratus ventralis*) (fat content 27.5 g 100 g^−1^). After fat adjustment, NaCl was added and blended to the mince before vacuum packing and freezing at −18 °C until trials begun. The mince was thawed in a refrigerator (4 °C) overnight before meatballs were manufactured for the trial. Meatballs were either pan-fried or deep-fried in randomized order in refined rapeseed oil (Zeta, Stockholm, Sweden), containing 7.5 g 100 g^−1^ saturated fatty acids, 62.5 g 100 g^−1^ mono-unsaturated fatty acids, and 30 g 100 g^−1^ poly-unsaturated fatty acids, where the temperature in the pan was kept stable at 175 °C ± 1 °C controlled by a laser thermometer (IR-termometer Basetech IRT-350, Plano, Texas, USA) and the deep-frying temperature was kept stable at 160 °C ± 0.5 °C. When the meatballs had reached a 72 °C inner temperature, they were removed from heating and rested on paper towels until room temperature was reached. The meatballs were then stored in a refrigerator (4 °C) in sealed polyethylene bags for 1, 7, and 14 days before the level of lipid oxidation was measured using thiobarbituric reactive substances (TBARS).

### 2.4. Test of Antioxidants in Most Oxidized Pork Meatball

The 11 antioxidant powders and extracts were added in 100 and 200 mg kg^−1^ of gallic acid equivalents (GAE), respectively, based on their total phenols content (Table 1) to the most oxidized type of meatball from the above presented study, namely, pork with 2 g 100 g^−1^ NaCl and 20 g 100 g^−1^ fat (weight by weight, which was deep-fried. In order to incorporate the antioxidants into the meat batters, 90 g of mince was mixed by hand with 9 mL of antioxidant solution until all fluid had been absorbed by the mince. Meatballs were hand-rolled and weighed in triplicates to a standardized weight of 16.5 g ± 0.1 g before they were deep-fried as mentioned above at 160 °C until the inner temperature reached 72 °C. The meatballs were then let to rest on paper towels until room temperature was reached. After 14 days in the refrigerator (4 °C), the level of lipid oxidation was measured using thiobarbituric reactive substances (TBARS).

### 2.5. Thiobarbituric Reactive Substances (TBARS)

The lipid oxidation was measured in triplicates for each meatball using TBARS based on the method of Buege and Aust [14], where the prepared TBA reagent consisted of 15 g 100 mL^−1^ TCA and 0.75 g 100 mL^−1^ TBA in 0.25 M HCl. Samples from meatballs were weighed to 6 ± 0.1 g and crushed by mortar and pestle prior to being added to a 50-mL falcon tube containing 22.5 mL Milli-Q H_2_O together with 1.5 mL of 10 g 100 mL^−1^ TCA solution. The samples were then vortexed for 60 s prior to heating at 40 °C for 5 min to precipitate proteins, before 2 mL 96 mL 100 mL^−1^ ethanol was added to solubilize fats. The mixtures were then filtered through a filter paper (Munktell, grade 1F) to obtain a clear filtrate for TBARS analysis. In total, 2.5 mL of the TBA reagent was added to 0.5 mL of the filtrates, which were then heated to 90 °C for 10 min before being cooled in tap water to end the reaction. All samples were centrifuged at 3600 g for 20 min (Allegra^®^ X-15R, Beckman Coulter, Brea, CA, USA) before the absorbance of the supernatants was measured at 534 nm and 600 nm, respectively (Varian Cary^®^ 50 UV-Vis Spectrophotometer, Agilent, Santa Clara, CA, USA). If the coefficient of variation (cv%) between the sample replicates exceeded 10%, the analyses were re-run. Results are reported as µM malondialdehyde (MDA) g^−1^ meatball for meatball parameter evaluations, and as a percentage oxidation of the blank sample (without added antioxidants) for antioxidant evaluation.

### 2.6. Statistical Analyses

Statistical analyses were carried out using SPSS Statistics 25.0 (IBM Corp. Released 2017. IBM SPSS Statistics for Windows, Version 25.0. Armonk, NY, USA) and R for Windows GUI front-end version 3.5.3 (R version 3.5.2 (2018-12-20)—“Eggshell Igloo” Copyright © 2018 The R Foundation for Statistical Computing Platform: x86_64-w64-mingw32/x64 (64-bit). Box-Cox transformations were used to achieve normal distribution of data. A univariate general linear model (GLM) was performed on logarithmic values of oxidation data in both the preliminary trial and antioxidant study in order to ensure normal distribution of samples (skewness and kurtosis with maximum values of ±1.96 [15]). In the preliminary trial, meat type, salt content, fat content, cooking type, and storage time were all fixed factors tested in a full-factorial GLM model on the dependent variable TBAR (µM MDA g^−1^ meatball). In the antioxidant study, species and concentrations of antioxidants were fixed factors in the full-factorial model of the dependent variable percent lipid oxidation of meatballs compared to meatballs without antioxidants. In both trials, meatball samples were made in triplicates of which TBARS was measured in 3 technical replicates; statistical analyses were conducted on an average of the results of the meatball triplicates. Post-hoc tests were performed using the Scheffe method. A Pearson correlation analysis was conducted using SPSS on previous antioxidant activity data for the samples included in this study together with the lipid oxidation inhibiting capacity to study if, and if so, which antioxidant mode of action gave rise to the lipid oxidation inhibiting capacity.

## 3. Results

### 3.1. Preliminary Trial for Meatball Model Selection

Lipid oxidation levels of meatballs differing in composition, cooking method, and time of storage are presented in Figure 2. Statistical results indicate that meat type, storage time, cooking type, and salt content all had significant effects on the lipid oxidation (*p* < 0.001) as did the fat content (*p* < 0.01). Nearly all interactions had significant effects on lipid oxidation (*p* < 0.05) except for interactions between: Salt × cooking, meat × salt × storage, meat × salt × fat × cooking, and meat × salt × fat × storage (Table A1). However, the partial Eta squared (η^2^) showed the effect sizes of most interactions were minor. The factors affecting lipid oxidation the most were meat type and storage time (*p* < 0.001) (Table A1), where pork oxidized more than beef, and where oxidation increased with longer storage times for both meat types, as is shown in the interaction between meat type and storage time (Figure 3a). There was an interaction between meat type and fat content (Figure 3b), where beef with 20 g 100 g^−1^ fat oxidized less than with 10 g 100 g^−1^ fat, and the opposite was shown in pork meat. The combination of parameters that oxidized the most contained pork meat, 20 g 100 g^−1^ fat, and 2 g 100 g^−1^ salt; was deep-fried; and was stored for 14 days (Figure 2) (Table A2).

### 3.2. Inclusion of Antioxidants in Most Oxidized Meatball

All meatball samples were evaluated for their TBAR substances (malondialdehyde (MDA) µM g^−1^) in triplicates after 14 days of storage where the results were calculated as a percentage of oxidation compared to samples without added antioxidants ((µM MDA g^−1^ meatball with plant material/µM MDA g^−1^ meatball without antioxidant) * 100) (Figure 4). Statistical results indicate that antioxidant species, concentrations, and the interaction of both significantly affected the level of lipid oxidation compared to the meatball without added antioxidants (*p* < 0.001) (Figure 4) of which the antioxidant species was shown to have the largest effect size on the level of lipid oxidation (Table A3). The summer savory powder (*Satureja hortensis* L., SS) at 200 mg kg^−1^ and 100 mg kg^−1^, water extracted sea buckthorn (*Hippophae rhamnoides* L., SBTH2O) at 100 mg kg^−1^, and olive polyphenols (*Olea europaea* L., OPP) at 200 mg kg^−1^ were the most efficient antioxidants, lowering lipid oxidation to 13.8%, 21.8%, 22.9%, and 26.1%, respectively (Table A4), compared to the meatball with no added antioxidants. There were no significant correlations (Pearson) between the total phenols content (GAE mg mL^−1^) and the lipid oxidation inhibition capacity of the antioxidants (Table A5).

## 4. Discussion

This study was divided into two parts, where the aim of the first part was to select a meatball model prone to oxidize. The aim of the second part of this work was to include and test various types of natural antioxidants for lipid oxidation inhibiting purposes. Both parts of the study will be discussed separately hereunder.

### 4.1. Preliminary Trial for Meatball Model Selection

We found that meat type was the most important factor affecting lipid oxidation (Table A1) where the pork meatballs overall were significantly (*p* < 0.001) more prone to oxidize than those of beef meat (Figure 3). This was not according to our expectations since pork meat contains less myoglobin (2 mg g^−1^) than beef meat (8 mg g^−1^) [16], and since the heme-iron content in meat is commonly hypothesized to be one of the significant substances in red and processed meat inducing carcinogenesis due to its involvement in mutagenic nitroso-compounds (NOCs) and the production of reactive oxygen species (ROS), producing lipid oxidation secondary products, such as malondialdehyde (MDA) [3,17,18]. However, a plausible explanation for the obtained results of the meatballs made of beef or pork meat could be differences in the levels of polyunsaturated fatty acids (PUFA) frequently involved in lipid oxidation. Pork meat has considerably higher amounts of linoleic acid (18:2) in both adipose tissue (14.3 g 100 g^−1^) and muscle tissue (14.2 g 100 g^−1^) than beef adipose tissue (1.1 g 100 g^−1^) and muscle tissue (2.4 g 100 g^−1^) [19]. This difference in linoleic acid (18:2) content could be of importance, since the oxidation in linoleic acid occurs 10 times faster than in oleic acid (18:1), and 20 to 30 times faster than linolenic acid (18:3) [17].

The meatball that oxidized the most contained pork, 20 g 100 g^−1^ fat, and 2 g 100 g^−1^ salt, and had been deep-fried and stored for 14 days (Figure 2). This meatball type was then chosen for further analysis, although it did not statistically differ from other meatballs (Table A2), since the aim remained to find the meatball type with the most oxidizing combinations of parameters. Interestingly, the beef meatballs that contained 10 g 100 g^−1^ fat and 2 g 100 g^−1^ salt that had been pan-fried and stored for one and 14 days oxidized more than the other combinations of beef meatballs (Figure 2), which could be an explanation for the antagonistic interaction between fat content and meat type (Figure 3b). Initially, we hypothesized that the higher the fat and salt contents in the meatballs, the higher the level of lipid oxidation, but this was not shown to be the case in our study. Salt (NaCl) is generally known to be a pro-oxidant in meat and meat products [5]; however, salt contents over 3 g 100 g^−1^ have shown little to no pro-oxidant effect [17], which is in accordance with the findings in the present study (Figure 3).

Although we found significant effects of cooking types on the level of lipid oxidation (Table A1), both deep-frying and pan-frying are considered to belong to the dry heat cooking category [20]. Hence, the heat transfer could be considered to be equal between the cooking types in matters of changes in physiochemical properties. When meatballs were analyzed for fat loss during pan-frying, Granheimer [13] found that the beef meatballs had higher fat loss than those of pork. However, in our study, pan-fried beef meatballs with 10 g 100 g^−1^ fat and 2 g 100 g^−1^ salt, which had higher TBARS levels (Figure 3), instead gained fat. In the same study, beef meatballs were shown to gain more fat during deep-frying than pork meatballs, in which fat was lost [13]. Haak et al. [21] found that the fatty acid composition of pan-fried meats became similar to that of the culinary fat due to its fat uptake. This could then explain why the beef meatball oxidized more than the others since rapeseed oil contains 20.9 g 100 g^−1^ PUFA [22] and thus increases the susceptibility to oxidation compared to the other beef meatballs. That all meatballs oxidized more over time (*p* < 0.001) (Figure 3) was expected due to the chain reaction nature of the lipid oxidation reaction.

### 4.2. Test of Antioxidants in the Most Oxidized Pork Meatball

Synthetic antioxidants have typically been included into meat products to increase, e.g., shelf-life and nutritional value, and now the industry is demanding new natural sources of antioxidant ingredients [5]. Numerous trials and experiments have successfully been carried out in screening the lipid oxidation inhibiting capacity of extracts from plant materials in various meat products [7]. For instance, olive leaf extracts have previously shown to be a potent antioxidant in bovine and porcine muscle model systems [9], oregano and sage essential oils have significantly reduced levels of lipid oxidation [8], and summer savory has previously been shown to reduce lipid oxidation in pork meatballs significantly [23]. All samples tested in our study effectively inhibited lipid oxidation at both concentrations (*p* < 0.001) (Figure 4), where the summer savory (SS) powder was the most efficient at both concentrations, and statistically differed from all other samples at 200 mg kg^−1^ (Table A4). The statistical results showed that antioxidant species, concentration, and the interaction of both had significant effects on lipid oxidation (*p* < 0.001), where species had the largest effect size (Table A5). Overall, samples inhibited lipid oxidation more efficiently at 200 mg kg^−1^ than at 100 mg kg^−1^ (Figure 4) except for the sea buckthorn sample extracted with 50 mL 100 mL^−1^ ethanol (SBT), which was significantly more effective at 100 mg kg^−1^, and the sea buckthorn sample extracted with H_2_O (SBTH2O), which showed tendencies to be more efficient at 100 mg kg^−1^. Radenkovs et al. [24] previously attributed this phenomenon to the phenol composition of each antioxidant, where reaction speed predominantly depends on each phenol’s chemical structure rather than its concentration. This explanation is further reinforced when interpreting the non-significant correlation analysis (Table A5) between the total phenols content and inhibition of lipid oxidation capacity. In further research, meatballs without and with antioxidants will be tested in vivo to evaluate potential changes in intestinal inflammatory reactions following a diet consisting of 20 g 100 g^−1^ of these meatballs.

## 5. Conclusions

The aim of the current study was to evaluate the lipid oxidation inhibiting capacity of natural antioxidants in a readily oxidized meat product. Various meatball properties were studied in order to find the combination that gave rise to the most lipid oxidation. The meatball most prone to oxidize was deep-fried, made of pork, contained 20 g 100 g^−1^ fat and 2 g 100 g^−1^ salt, and had been stored for 14 days. This meatball type was then manufactured without and with 11 different plant materials and extracts at two concentrations, 100 mg kg^−1^ and 200 mg kg^−1^ GAE, and was stored for 14 days. All samples inhibited lipid oxidation effectively in both tested concentrations, where the summer savory powder was the most efficient in both the 100 mg kg^−1^ and 200 mg kg^−1^ concentration, lowering lipid oxidation to 21.8% and 13.8%, respectively, compared to meatballs with no added antioxidants. Thus, antioxidant rich plant materials and extracts could efficiently prevent lipid oxidation in processed meat products, such as meatballs.

## Figures and Tables

**Figure 1 foods-08-00406-f001:**
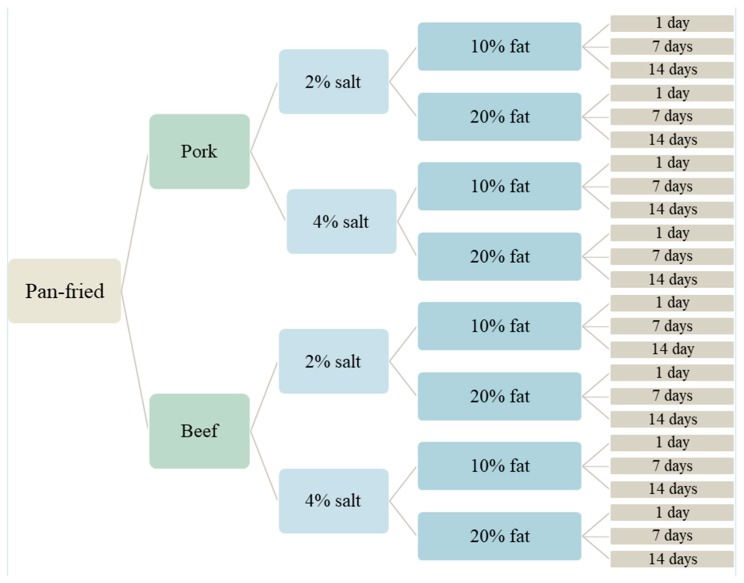
Scheme of the pan-fried meatball study set up as modified by Granheimer [13]. The same set up was used for the deep-fried meatball study.

**Figure 2 foods-08-00406-f002:**
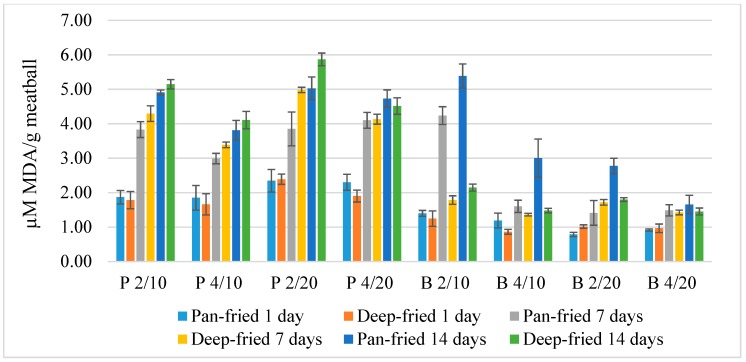
Lipid oxidation in model meatballs with differing parameters shown in µM malondialdehyde (MDA) g^−1^ meatball, where P = pork and B = beef meat. The numbers 2 and 4 correspond to the salt content in % and the numbers 10 and 20 correspond to the fat content in %. The standard deviation is shown by the error bars (*n* = 3).

**Figure 3 foods-08-00406-f003:**
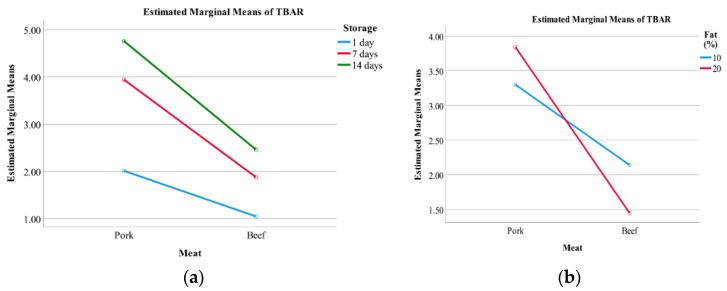
Main effect plots (estimated marginal means) of TBARS (µM malondialdehyde g^−1^ meatball) of (**a**) the interaction between meat type and storage times and (**b**) the interaction between meat type and fat contents (%).

**Figure 4 foods-08-00406-f004:**
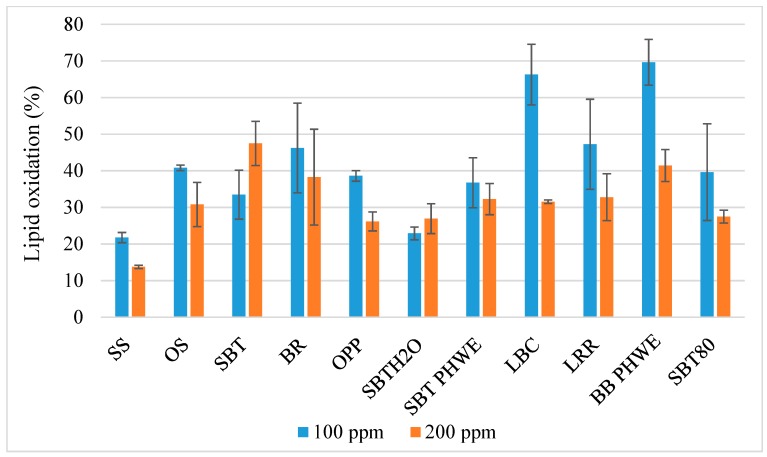
Lipid oxidation in the meatball type most prone to oxidize (pork with 2% NaCl, 20% fat (w/w), deep-fried), with different antioxidants shown as a percentage of oxidation compared to the meatball without added antioxidants at two concentrations. SS = Summer Savory, OS = Onion skin, SBT = Sea buckthorn leaves, BR = Beetroot leaves, OPP = Olive polyphenols, SBTH2O = water extracted sea buckthorn leaves and sprouts, SBT PHWE = Sea buckthorn leaves and sprouts-pressurized hot water extraction, LBC = Lyophilized black currant leaves, LRR = Lyophilized rhubarb root, BB PHWE = Bilberry leaves-pressurized hot water extraction, and SBT80 = ethanol (80%) extracted sea buckthorn leaves and sprouts. The standard deviation is shown by the error bars (*n* = 3).

**Table 1 foods-08-00406-t001:** Samples of antioxidant plant materials and extraction methods used.

Plant Material	Latin Name	Cultivar	Country	Extraction Solution	Abbreviation	Extraction Method	GAE mg mL^−1^ Extract
Lyophilized Sea buckthorn leaves	*Hippophae rhamnoides* L.	’Botnia Guldklimp’	Finland	Pressurized hot water	SBTPHWE	Section 2.1	7.0
Lyophilized Bilberry leaves	*Vaccinium myrtillus* L.	Native stands	Finland	Pressurized hot water	BBPHWE	Section 2.1	11.6
Sea buckthorn leaves and sprouts	*Hippophae rhamnoides* L.	Mix of ‘Botaņičeskaja Ļubiteļskaja’ and ‘Prozračnaja’	Latvia	80% Ethanol	SBT80	Gornas et al. [12]	13.2
Sea buckthorn leaves and sprouts	*Hippophae rhamnoides* L.	Mix of ‘Botaņičeskaja Ļubiteļskaja’ and ‘Prozračnaja’	Latvia	H_2_O	SBTH_2_O	Modified from Gornas et al. [12]	9.2
Summer savory leaves	*Satureja hortensis* L.	Seed origin: Hild Samen	Denmark	Non-extracted	SS	Non-extracted	12.0
Sea buckthorn leaves	*Hippophae rhamnoides* L.	’Finskaja’	Sweden	50% ethanol	SBT	Burri et al. [11]	8.8
Olive Polyphenols-Phenoliv	*Olea europaea* L.	Phenoliv™	Sweden	50% ethanol	OS	Burri et al. [11]	3.8
Onion skin	*Allium cepa* L.	’Donna’	Sweden	50% ethanol	OPP	Burri et al. [11]	3.0
Beetroot leaves	*Beta vulgaris* subsp. *vulgaris*	’Action’	Sweden	50% ethanol	BR	Burri et al. [11]	1.0
Lyophilized rhubarb root	*Rheum rhabarbarum* L.	’Victoria’	Estonia	20% ethanol	LRR	Section 2.1	18.1
Lyophilized black currant leaves	*Ribes nigrum* L.	’Pamjat Vavilova’	Estonia	20% ethanol	LBC	Section 2.1	10.1

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
