# Peer review of "Lipid Oxidation Inhibition Capacity of 11 Plant Materials and Extracts Evaluated in Highly Oxidised Cooked Meatballs"

_foods, 2019, doi:10.3390/foods8090406_

Round 1

Reviewer 1 Report

The paper is well written & describes the lipid oxidation inhibition capacity of 11 plant materials.

The results are good & the analysis of results significant. Summer  savory freeze-dried powder was most efficient, lowering lipid oxidation to 13.8 and 21.8 % at 200  and 100 mg kg-1 concentration respectively, followed by a sea buckthorn extract, lowering lipid oxidation to 22.9 % at 100 mg kg-1, compared to the meatball without added antioxidants. The lipid 24 oxidation was thus successfully reduced using these natural antioxidants, a result which will be valuable to the food industry.

Author Response

Point 1. The paper is well written & describes the lipid oxidation inhibition capacity of 11 plant materials.

Response 1. The authors thank reviewer 1 for the kind words on the paper.  

Point 2. The results are good & the analysis of results significant. Summer  savory freeze-dried powder was most efficient, lowering lipid oxidation to 13.8 and 21.8 % at 200  and 100 mg kg-1 concentration respectively, followed by a sea buckthorn extract, lowering lipid oxidation to 22.9 % at 100 mg kg-1, compared to the meatball without added antioxidants. The lipid 24 oxidation was thus successfully reduced using these natural antioxidants, a result which will be valuable to the food industry.

Response 2. The authors appreciate that reviewer 1 agrees with the authors on the importance of this study for the food industry.

Point 3. Reviewer 1 expresses the need for extensive editing of English language and style.

Response 3. The manuscript has been corrected by a native English speaking colleague. 

Reviewer 2 Report

2. Materials and methods:

Authors tell that extracts used in the study were prepared in different countries. It is not clear whether the Authors of the manuscript (all from Sweden) conduct all the extractions by themselves. If  the extractions made in Estonia, Finland, Denmark and Latvia were actually made in partner Institutes of the project by the personel of that institute, the referee would have presumed that the authors would acknowledge the partnering organisations where the work has been actualised; and this would be also according to common codes of scientific cooperation. Of course, this is not necessary, if any of the named authors has accomplished the extractions by him/herself in those countries. 

M&M written text and Table 1 are a bit inconsistent, when estonian samples are named (in Table 1) Lyophilised ...., when  they were freeze-dried after extraction but the finnish samples processed same way after extraction were not named this way.

Latvian method is missing; authors could add the method description or mention it was conducted the same manner as in any of the other countries. As well, the swedish extraction experiment is not possible to repeat after the description given in the manuscript. 

In the format given to reviewers:
Table 1 columns should be wider so that each row is only one row (e.g. <0,001 is divided to two rows.)

Author Response

Point 1.

Materials and methods:

Authors tell that extracts used in the study were prepared in different countries. It is not clear whether the Authors of the manuscript (all from Sweden) conduct all the extractions by themselves. If  the extractions made in Estonia, Finland, Denmark and Latvia were actually made in partner Institutes of the project by the personel of that institute, the referee would have presumed that the authors would acknowledge the partnering organisations where the work has been actualised; and this would be also according to common codes of scientific cooperation. Of course, this is not necessary, if any of the named authors has accomplished the extractions by him/herself in those countries. 

Response 1. It is correct that the extractions were carried out in the partner institutes of our project. This manuscript is the third work package of the project called "Sustainable plant ingredients for healthier meat products - proof of concepts". In the second work package, 28 plant materials (including the 11 plant materials in the present manuscript) were tested in a meat model system, in which all institutes and colleagues were included as co-authors in the manuscript (as agreed within the project). Therefore, these were not named in the present manuscript. However, reviewer 2 is entirely correct in that these institutes and colleagues should be acknowledged in the present manuscript as well, why we chose to acknowledge them under the "acknowledgement" section. 

Point 2. M&M written text and Table 1 are a bit inconsistent, when estonian samples are named (in Table 1) Lyophilised ...., when  they were freeze-dried after extraction but the finnish samples processed same way after extraction were not named this way.

Response 2. Reviewer 2 is correct. This is inconsistent and the Finnish samples have now been renamed in Table 1 and "freeze-dried" has now been changed into "lyophilised" in the text. 

Point 3. Latvian method is missing; authors could add the method description or mention it was conducted the same manner as in any of the other countries. As well, the swedish extraction experiment is not possible to repeat after the description given in the manuscript. 

Response 3. The Latvian and Swedish extraction methods have reference articles in Table 1 where the extraction methods could be studied. 

Point 4. In the format given to reviewers:
Table 1 columns should be wider so that each row is only one row (e.g. <0,001 is divided to two rows.)

Response 4. Reviewer 2 is correct and this was an unfortunate formatting mistake. Table 1 in the Appendix has now been enlarged to fit all columns. 

Point 5. Reviewer 2 expresses that English language and style are fine or needs minor spell checks.

Response 5. The manuscript has been read and corrected by a native English speaker.

Reviewer 3 Report

The paper entitled “Lipid oxidation inhibition capacity of 11 plant materials and extracts evaluated in a minced meat model product” is well written and of high interest for meat industry.

However, I have some questions and I will appreciate some clarifications to be made.

The title is very general, please highlight that the model was developed for highly oxidized cooked meatballs.

Abstract

Line 18 deepfried meatballs

Line 21 14 days of storage

Line 21 delete triplicates

Line 22 complete the sentence, was the most efficient compound assayed Page numbers before lines 81 and 83 seem wrong.

Introduction I know cancer related with meat is of great importance nowadays, however the use of this antioxidants will be also of importance for other reasons, such as enlargement of shelf life, nothing have been included in the introduction about it.

I have some concerns about the use of the word natural. Since some of the compounds used were “artificially” extracted, is it appropriate the use of the word natural?

Material and methods
Why were the antioxidant used in different formats? Why summer Savory was not extracted for example? How did you take this decision?

In the same line? The final amount of water incorporated to the meatballs was the same in all the cases? Finally The solvents used were food grade?

Line 78 please describe if it is MilliQ, distillated or tap water,
Delete the last sentence, it is explained below, and the parentheses around Folin-Ciocalteou.

Line 92 could you provide the fat composition of the muscles? Or the amount of added fat included in each of them? Since the composition of the fat in the muscle and the added one is different, it may affect the results.
Also it catch my attention why did you decided to store your meatballs once cooked? Is this common? It will not be more common to store the raw meatball?

The meatballs were only done with meat did you used the other common ingredients such us breadcrumbs?

The storage conditions have not been described.
Line 110 include the real units like in the abstract mg per kg of gallic acid equivalents (FAE)

Line 113 Here is not clear if the meatballs were done in triplicates.

Line 119 Since is not described above here we don´t know if the triplicates were 3 independent meatballs or three subsamples of the same meatball.

Line 121 was it weighted before crush? Wouldn’t it be more accurate to weight the exact amount directly once crushed in the falcon tube?

The section Statistical analysis must be widely improved, you must present individually the model with the fixed effect used per each of the analysis (preliminary and antioxidant efficiency), also if did you used the repeated stamen or similar. The way the statistics are performed will importantly affect the soundness and reliability of your results.

Line 142 Describe why did you use the Pearson correlation analysis for
Results

The title of the subsection is very general shouldn´t it be something like. Preliminary trial for meatball model selection?
Delete lines 145 to 147 this is M&M and describe what is presented in figure

3. Something like:
Lipid oxidation levels on meatballs differing in composition, cooking method and time of storage are presented in Figure 3.

Line 146 is Figure 2 mentioned somewhere before line 146 if not you should change the order of the figures.

I think that tables 1 and 2 are more important in the results than for example figures 2 why did you decided to include them as appendix and not near the text?

Line 147 Delete GLM-analysis here and elsewhere, this is the methodology, so start with Statistical results indicate…
Line 174 Incomplete title b)
Line 182 define better the units MDA per gram of what?
Line 182 to 183 Delete this is M&M
Line 183 Include here the equation of the calculation made.
Line 188 include the specie like in the other compounds.
Lines 206 to 210 is more an introduction than something that must be included in the discussion. Better describe that you perform to separated experiment and you are going to discuss both separately in two sections.
Line 211 Change the title in line with the one in the results
Line 220 Can be also related with differences in the composition and amount of extra-fat included in meatballs?

In general in the discussion modify and when you talk about weeks modify including days since your data is presented by day.

Line 225 Actually, if the group column presented in Table 2 indicates that same letters indicate not significant differences, in that case this did not differ from pork at 2 10 deep fried, Pork 2 20 pan fried Pork 2 20 deep fried at 7 days and Pork 2 10 pan fried at 14 days. So its not correct to said that it was the most oxidized. Same in some other descriptions. Please modify here and elsewhere, bee careful when talking about differences and when stating that something is the most, because this should mean that it is significantly higher compared with any other treatment.

Discussion
In general the discussion need a most deep review of bibliography, there is any other people using similar compounds in other models or not? If not it should be also stated. There is any other person that have investigated SS as a food antioxidant with similar results?.

The size of the effect in GLM is usually given by eta partial square in GLM using SPSS, how did you calculate the importance of the effect?

Tables in general are not completed and they should be widely improved.
Please describe any abbreviation CL, df, SE.

Describes what group means? O

Is SE the standard error of the mean? describe it please.

Also for this parameters such as df and SE that are not modified just stated in the comments above the table that the values for them were:____ and ____ you don´t need to use a whole column for this. Clear tables with only the most important information help with the interpretation the data

In the references please provide the abbreviated journal in all cases
Line 340 correct paragraph format.
Check the format of reference 19
